# Open Vocabulary Learning on Source Code with a Graph–Structured Cache

## Abstract

Machine learning models that take computer program source code as input typically use Natural Language Processing (NLP) techniques. However, a major challenge is that code is written using an open, rapidly changing vocabulary due to, e.g., the coinage of new variable and method names. Reasoning over such a vocabulary is not something for which most NLP methods are designed. We introduce a Graph–Structured Cache to address this problem; this cache contains a node for each new word the model encounters with edges connecting each word to its occurrences in the code. We find that combining this graph–structured cache strategy with recent Graph–Neural–Network–based models for supervised learning on code improves the models' performance on a code completion task and a variable naming task — with over 100% relative improvement on the latter — at the cost of a moderate increase in computation time.

## 1 Introduction

Computer program source code is an abundant and accessible form of data from which machine learning algorithms could learn to perform many useful software development tasks, including variable name suggestion, code completion, bug finding, security vulnerability identification, code quality assessment, or automated test generation. But despite the similarities between natural language and source code, deep learning methods for Natural Language Processing (NLP) have not been straightforward to apply to learning problems on source code (Allamanis et al., 2017).

There are many reasons for this, but two central ones are:

1. *Code's syntactic structure is unlike natural language.* While code contains natural language words and phrases in order to be human–readable, code is not meant to be a read like a natural language text. Code is written in a rigid syntax with delimiters that may open and close dozens of lines apart; it consists in great part of references to faraway lines and different files; and it describes computations that proceed in an order often quite distinct from its written order.

2. *Code is written using an open vocabulary.* Natural language is mostly composed of words from a large but closed (a.k.a. fixed–size and unchanging) vocabulary. Standard NLP methods can thus perform well by fixing a large vocabulary of words before training, and labeling the few words they encounter outside this vocabulary as "unknown". But in code every new variable, class, or method declared requires a name, and this abundance of names leads to the use of many obscure words: abbreviations, brand names, technical terms, etc.[1] A model must be able to reason about these newly–coined words to understand code.

The second of these issues is significant. To give one indication: 28% of variable names contain out–of–vocabulary words in the test set we use in our experiments below. But more broadly, the open vocabulary issue in code is an acute example of a fundamental challenge in machine learning:

---

[1]We use the terminology that a *name* in source code is a sequence of *words*, split on `CamelCase` or `snake_case`. E.g. the method name `addItemToList` is composed of the words `add`, `item`, `to`, and `list`.
We also use the term *variable* in its slightly broader sense to refer to any user–named language construct, including function parameter, method, class, and field names, in addition to declared variable names.

how to build models that can reason over unbounded domains of entities, sometimes called "open–set" learning. Despite this, the open vocabulary issue in source code has received relatively little attention in prior work.

The first issue, in contrast, has been the focus of much prior work. A common strategy in these works is to represent source code as an Abstract Syntax Tree (AST) rather than as linear text. Once in this graph–structured format, code can be passed as input to models like Recursive Neural Networks or Graph Neural Networks (GNNs) that can, in principle, exploit the relational structure of their inputs and avoid the difficulties of reading code in linear order (Allamanis et al., 2018).

**Our contribution:** In this paper we extend such AST–based models for source code in order to address the open vocabulary issue. We do so by introducing a Graph–Structured Cache (GSC) to handle out–of–vocabulary words. The GSC represents vocabulary words as additional nodes in the AST as they are encountered and connects them with the edges to where they are used in the code. We then process the AST+GSC with a GNN to produce outputs. See Figure 1.

We empirically evaluated the utility of a Graph–Structured Cache on two tasks: a code completion (a.k.a. fill–in–the–blank) task and a variable naming task. We found that using a GSC improved performance on both tasks at the cost of an approximately 30% increase in training time. More precisely: even when using hyperparameters optimized for the baseline model, adding a GSC to a baseline model improved its accuracy by at least 7% on the fill–in–the–blank task and 103% on the variable naming task. We also report a number of ablation results in which we carefully demonstrate the relative importance of each model component to a model's performance.

## 2 PRIOR WORK

### REPRESENTING CODE AS A GRAPH

Given their prominence in the study of programming languages, Abstract Syntax Trees (ASTs) and parse trees are a natural choice for representing code and have been used extensively. Often models that operate on source code consume ASTs by linearizing them (usually with a depth–first traversal) as in Amodio et al. (2017), Liu et al. (2017), or Li et al. (2017) or by using AST paths as input features as in Alon et al. (2018), but they can also be processed by deep learning models that take graphs as input, as in White et al. (2016) and Chen et al. (2018) who use Recursive Neural Networks (RveNNs) (Goller & Kuchler, 1996) on ASTs. RveNNs are models that operate on tree–topology graphs, and have been used extensively for language modeling (Socher et al., 2013) and on domains similar to source code, like mathematical expressions (Zaremba et al., 2014; Arabshahi et al., 2018). They can be considered a special case of Message Passing Neural Networks (MPNNs) in the framework of Gilmer et al. (2017): in this analogy RveNNs are to Belief Propagation as MPNNs are to Loopy Belief Propagation. They can also be considered a special case of Graph Networks in the framework of Battaglia et al. (2018). ASTs also serve as a natural basis for models that generate code as output, as in Maddison & Tarlow (2014), Yin & Neubig (2017), Rabinovich et al. (2017), Chen et al. (2018), and Brockschmidt et al. (2018).

Data–flow graphs are another type of graphical representation of source code with a long history (Krinke, 2001), and they have occasionally been used to featurize source code for machine learning (Chae et al., 2017).

Most closely related to our work is the work of Allamanis et al. (2018), on which our model is heavily based. Allamanis et al. (2018) combine the data–flow graph and AST representation strategies for source code by representing code as an AST augmented with extra labeled edges indicating semantic information like data– and control–flow between variables. These augmentations yield a directed multigraph rather than just a tree,[2] so in Allamanis et al. (2018) a variety of MPNN called a Gated Graph Neural Network (GGNN) (Li et al., 2016) is used to consume the Augmented AST and produce an output for a supervised learning task.

Graph-based models that are not based on ASTs are also sometimes used for analyzing source code, like Conditional Random Fields for joint variable name prediction in (Raychev et al., 2015).

---

[2]This multigraph was referred to as a Program Graph in Allamanis et al. (2017) and is called an Augmented AST herein.

REASONING ABOUT OPEN SETS

The question of how to gracefully reason over an open vocabulary is longstanding in NLP. Character–level embeddings are a typical way deep learning models handle this issue, whether used on their own as in Kim et al. (2016), or in conjunction with word–level embedding Recurrent Neural Networks (RNNs) as in Luong & Manning (2016), or in conjunction with an $n$–gram model as in Bojanowski et al. (2017). Another approach is to learn new word embeddings on–the–fly from context (Kobayashi et al., 2016). Caching novel words, as we do in our model, is yet another strategy (Grave et al., 2017) and has been used to augment N–gram models for analyzing source code (Hellendoorn & Devanbu, 2017).

In terms of producing outputs over variable–sized input and outputs, also known as open–set learning, attention-based pointer mechanisms were introduced in Vinyals et al. (2015) and have been used for tasks on code, e.g. in Bhoopchand et al. (2016). Such methods have been used to great effect in NLP in e.g. Gulcehre et al. (2016) and Merity et al. (2017). The latter's pointer sentinel mixture model is the direct inspiration for the readout function we use in the Variable Naming task below.

Using graphs to represent arbitrary collections of entities and their relationships for processing by deep networks has been widely used (Johnson, 2017; Bansal et al., 2017; Pham et al., 2018; Lu et al., 2017), but to our knowledge we are the first to use a graph–building strategy for reasoning (at train *and* test time) about an open vocabulary of words.

## 3 PRELIMINARIES

### 3.1 ABSTRACT SYNTAX TREES

An Abstract Syntax Tree (AST) is a graph — specifically an ordered tree with labeled nodes — that is a representation of some written computer source code. There is a 1–to–1 relationship between source code and an AST of that source code, modulo comments and whitespace in the written source code.

Typically the leaves of an AST correspond to the tokens written in the source code, like variable and method names, while the non–leaf nodes represent syntactic language constructs like function calls or class definitions. The specific node labels and construction rules of ASTs can differ between or within languages. The first step in Figure 1 shows an example.

### 3.2 GRAPH NEURAL NETWORKS

The term Graph Neural Network (GNN) refers to any deep, differentiable model that takes graphs as input. Many GNNs have been presented in the literature, and several nomenclatures have been proposed for describing the computations they perform, in particular in Gilmer et al. (2017) and Battaglia et al. (2018). Here we give a brief recapitulation of supervised learning with GNNs using the Message Passing Neural Network framework from Gilmer et al. (2017).

A GNN is trained using pairs $(G, y)$ where $G = (V, E)$ is a graph defined by its vertices $V$ and edges $E$, and $y$ is a label. $y$ can be any sort of mathematical object: scalar, vector, another graph, etc. In the most general case, each graph in the dataset can be a directed multigraph, each with a different number of nodes and different connectivity. In each graph, each vertex $v \in V$ has associated features $\boldsymbol{x}_v$, and each edge $(v, w) \in E$ has features $\boldsymbol{e}_{vw}$.

A GNN produces a prediction $\hat{y}$ for the label $y$ of a graph $G = (V, E)$ by the following procedure:

1. A function $S$ is used to initialize a hidden state vector $\boldsymbol{h}_v^0$ for each vertex $v \in V$ as a function of the vertex's features (e.g., if the $\boldsymbol{x}_v$ are words, $S$ could be a word embedding function):
$$\boldsymbol{h}_v^0 = S(\boldsymbol{x}_v)$$

2. For each round $t$ out of $T$ total rounds:
   (a) Each vertex $v \in V$ receives the vector $\boldsymbol{m}_v^{t+1}$, which is the sum of "messages" from its neighbors, each produced by a function $M_t$:
   $$\boldsymbol{m}_v^{t+1} = \sum_{w \in \text{neighbors of } v} M_t(\boldsymbol{h}_v^t, \boldsymbol{h}_w^t, \boldsymbol{e}_{vw}).$$

    (b) Each vertex $v \in V$ updates its hidden state based on the message it received via a function $U_t$:

$$\boldsymbol{h}_v^{t+1} = U_t(\boldsymbol{h}_v^t, \boldsymbol{m}_v^{t+1}).$$

3. A function $R$, the "readout function", produces a prediction based on the hidden states generated during the message passing (usually just those at from time $T$):

$$\hat{y} = R(\{\boldsymbol{h}_v^t | v \in V, t \in 1, ..., T\}).$$

GNNs differ in how they implement $S$, $M_t$, $U_t$, and $R$. But all these functions are differentiable and most are parameterized, so the model is trainable via stochastic gradient descent of a loss function on $y$ and $\hat{y}$.

## 4    MODEL

Our model consumes an input instance of source code and produces an output for a supervised learning task via the following five steps, sketched in Figure 1:

1. Parse the source code (snippet, file, repository, version control history, etc.) into an Abstract Syntax Tree.

2. Add edges of varying types (details in Appendix Table 8) to this AST that represent semantic information like data– and control– flow, in the spirit of Allamanis et al. (2018). Also add the reversed version of all edges with their own edge type. This results in a directed multigraph called an Augmented AST.

3. Further augment the Augmented AST by adding a Graph–Structured Cache. That is, add a node to the Augmented AST for each vocabulary word encountered in the input instance. Then connect each such "cache node" with an edge (of edge type `WORD_USE`) to all variables whose names contain its word.

4. Vectorize the Augmented AST + GSC graph into a form suitable for a GNN. (I.e. perform Step 1 from Section 3.2.) Each AST node that doesn't represent a variable is vectorized as a learned embedding of the language construct it represents, e.g. `Parameter`, `Method Declaration`, etc. Each cache node and each node that represents a variable is vectorized as a learned linear map of the concatenation of a type embedding and a name embedding. The name embedding is a Character–Level Convolutional Neural Network (CharCNN) (Zhang et al., 2015) embedding of the word/name the node contains. The type embedding is a learned embedding of the name of the Java type of the token it contains, e.g. `int`, a user–defined class, etc., with cache nodes having their own unique `Cacher Node` type.

5. Process the graph with a GNN, as per Section 3.2. (I.e. perform Steps 2 and 3 from Section 3.2.) The readout functions differ depending on the task and are described in the Experiments section below.

Our main contribution to previous works is the addition of Step 3, the Graph–Structured Cache step. The combination of relational information from the cache nodes' connections and lexical information from these nodes' CharCNN embeddings allows the model to, in principle, flexibly reason about words it never saw during training, but also recognize words it did. E.g. it could potentially see a class named "`getGuavaDictionary`" and a variable named "`guava_dict`" and both (a) utilize the fact that the word "guava" is common to both names despite having never seen this word before, and (b) exploit learned representations for words like "get", "dictionary", and "dict" that it has seen during training.

## 5    EXPERIMENTS

We evaluated our model, described in Section 4, on two supervised tasks: a Fill–In–The–Blank task and a Variable Naming task. For each task, we compare our model to others that differ in how they parse the code and how they treat the words they encounter. Table 1 details the different variations of the procedure in Section 4 against which we compare our model.

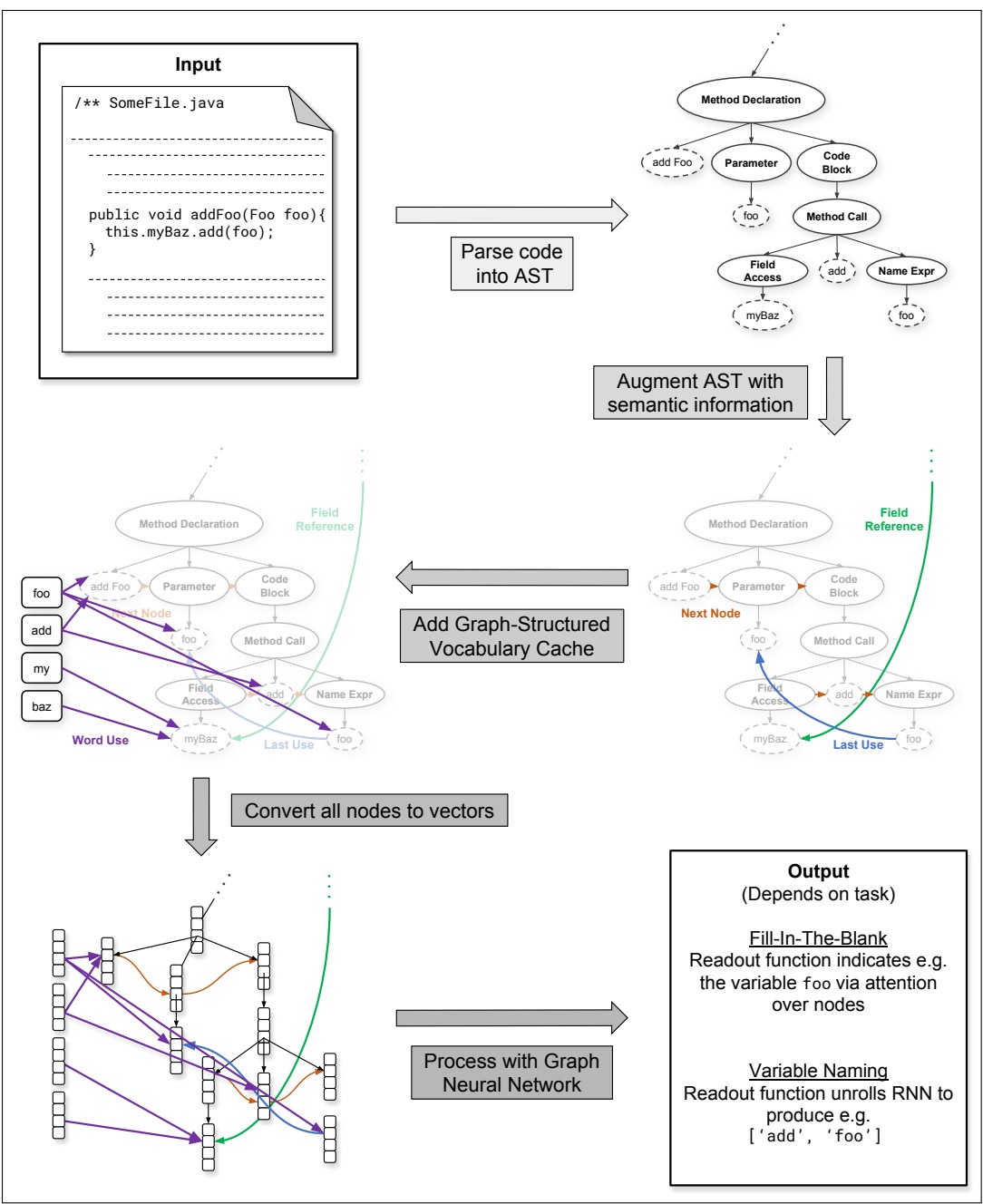

Figure 1: Our model's procedure for consuming a single input instance of source code and producing an output for a supervised learning task.

Code to reproduce all experiments is available online.[3] [4]

## 5.1 DATA AND IMPLEMENTATION DETAILS

We chose to use Java source code as the data for our experiments as it is among the most popular programming languages in use today (TIOBE, 2018; Github, 2017). To construct our dataset, we

---

[3]URL redacted to preserve review blind

[4]URL redacted to preserve review blind

Table 1: Nomenclature used in Experiments section. Each abbreviation describes a tweak/ablation to our full model as presented in Section 4. Using this nomenclature, our full model as described in Section 4 and shown in Figure 1 would be a "AugAST–GSC" model.

| | **Abbreviation** | **Meaning** |
|---|---|---|
| **Code Representation** | AST | Skips Step 2 in Section 4. |
| | AugAST | Performs Step 2 in Section 4. |
| **Vocab Strategies** | Closed Vocab | Skips Step 3 in Section 4, and instead maintains word–embedding vectors for words in a closed vocabulary. In Step 4, name embeddings for nodes representing variables are produced by taking the mean of the embeddings of the words in the variable's name. Words outside this model's closed vocabulary are labeled as <UNK>. This is the strategy used in Allamanis et al. (2018). |
| | CharCNN | Skips Step 3 in Section 4. |
| | Pointer Sentinel | Follows Steps 3 and 4 as described in Section 4, except it doesn't add edges connecting cache nodes to the nodes where their word is used. In the Variable Naming task, this is equivalent to using the Pointer Sentinel Mixture Model of Merity et al. (2017) to produce outputs. |
| | GSC | Follows Steps 3 and 4 as described in Section 4. |
| **Graph Neural Network** | GGNN | Performs Step 5 in Section 4 using the Gated Graph Neural Network of Li et al. (2016). |
| | DTNN | Performs Step 5 in Section 4 using the Deep Tensor Neural Network of Schütt et al. (2017). |
| | RGCN | Performs Step 5 in Section 4 using the Relational Graph Convolutional Network of Schlichtkrull et al. (2017). |

randomly selected 18 of the 100 most popular Java repos from the Maven repository[5] to serve as training data. (See Appendix Table 7 for the list.) Together these repositories contain about 500,000 non–empty, non–comment lines of code. We checked for excessive code duplication in our dataset (Lopes et al., 2017) using CPD[6] and found only about 7% of the lines to be contiguous, duplicated code blocks containing more than 150 tokens.

We randomly chose 3 of these repositories to sequester as an "Unseen Repos" test set. We then separated out 15% of the files in the remaining 15 repositories to serve as our "Seen Repos" test set. The remaining files served as our training set, from which we separated 15% of the datapoints to act as a validation set.

Our data preprocessor builds on top of the open–source Javaparser[7] library to generate ASTs of our source code and then augment the ASTs with the edges described in Appendix Table 8. We used Apache MXNet[8] as our deep learning framework. All hidden states in the GNN contained 64 units; all GNNs ran for 8 rounds of message passing; all models were optimized using the Adam optimizer (Kingma & Ba, 2015); all inputs to the GNNs were truncated to a maximum size of 500 nodes centered on the <FILL-IN-THE-BLANK> or <NAME-ME> tokens. About 53% of input graphs were larger than 500 nodes before truncation. The only regularization we used was early stopping — early in our experiments we briefly tried $L_2$ and dropout regularization, but saw no effects.

---

[5] https://mvnrepository.com/

[6] https://pmd.github.io/latest/pmd_userdocs_cpd.html

[7] https://javaparser.org/

[8] https://mxnet.apache.org/

We performed only a moderate amount of hyperparameter optimization, but all of it was done on the baseline models to avoid biasing our results in favor of our model. Specifically, we tuned all hyperparameters on the Closed Vocab baseline model, and also did a small amount of extra learning rate exploration for the Pointer Sentinel baseline model to try to maximize its performance.

## 5.2 The Fill–In–The–Blank Task

In this task we randomly selected a single usage of a variable in some source code, replaced it with a `<FILL-IN-THE-BLANK>` token, and then asked the model to predict what variable should have been there. An example instance from our dataset is shown in Figure 2.

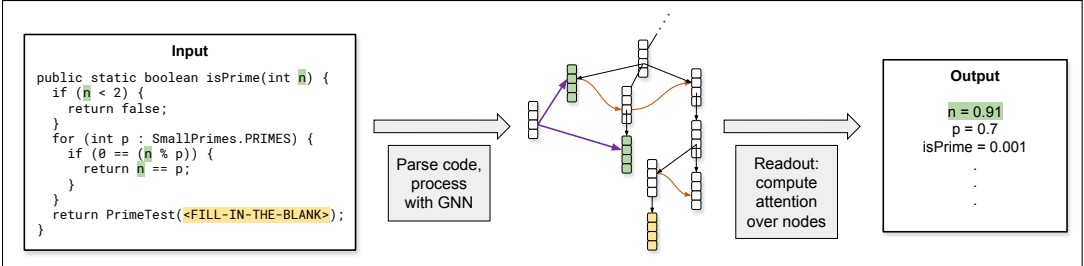

Figure 2: Example of a model's procedure for completing the Fill–In–The–Blank task. Each Fill–In–The–Blank instance is created by replacing a single usage of a variable (n, in this example) with the special token `<FILL-IN-THE-BLANK>`. The model then processes the code as depicted in Figure 1. To produce outputs, the model's readout function computes a soft–attention weighting over all nodes in the graph; the model's output is the variable at the node on which it places maximal attention. In this example, if the model put maximal attention weighting on any of the green–highlighted variables, this would be a correct output. If maximal attention is placed on any other node, it would be an incorrect output. Only in–scope usages of a variable are counted as correct.

The models indicate their prediction for what variable should go in the blank by pointing with neural attention over all the nodes in the AugAST. This means all training and test instances only considered cases where the obfuscated variable appears somewhere else in the code. Single uses are rare however, since in Java variables must be declared before they are used. It also means there are sometimes multiple usages of the same, correct variable to which a model can point to get the right answer. In our dataset 78% of variables were used more two times, and 33% were used more than four times.

The models compute the attention weightings $y_i$ for each Augmented AST node $i$ differently depending on the readout function of the GNN they use. Models using a GGNN as their GNN component, as all those in Table 2 do, compute the attention weightings as per Li et al. (2016):

$$\hat{y}_i = \sigma \left( f_1(\boldsymbol{h}_v^T, \boldsymbol{h}_v^0) \right) \odot f_2(\boldsymbol{h}_v^T),$$

where the $f$s are MLPs, $\boldsymbol{h}_v^t$ is the hidden state of node $v$ after $t$ message passing iterations, $\sigma$ is the sigmoid function, and $\odot$ is elementwise multiplication. The DTNN and RGCN GNNs compute the attention weightings as per Schütt et al. (2017):

$$\hat{y}_i = f(\boldsymbol{h}_v^T),$$

where $f$ is a single hidden layer MLP. The models were trained using a binary cross entropy loss computed across the nodes in the graph.

The performance of models using our GSC versus those using other methods is reported in Table 2. For context, a baseline strategy of random guessing among all variable nodes within an edge radius of 8 of the `<FILL-IN-THE-BLANK>` token achieves an accuracy of 0.22. We also compare the performance of different GNNs in Table 3.

## 5.3 The Variable Naming Task

In this task we replaced all usages of a name of a particular variable, method, class, or parameter in the code with the special token `<NAME-ME>`, and asked the model to produce the obfuscated

Table 2: Accuracy on the Fill–In–The–Blank task. Our model is the AugAST–GSC. The first number in each cell is the accuracy of the model, where a correct prediction is one in which the graph node that received the maximum attention weighting by the model contained the variable that was originally in the `<FILL-IN-THE-BLANK>` spot. The second, parenthetical numbers are the top–5 accuracies, i.e. whether the correct node was among those that received the 5 largest attentions weightings from the model. See Table 1 for explanations of the abbreviations. All models use Gated Graph Neural Networks as their GNN component.

|  |  | Closed Vocab | CharCNN | GSC |
|---|---|---|---|---|
| **Seen repos** | AST | 0.57 (0.83) | 0.60 (0.84) | 0.89 (0.96) |
|  | AugAST | 0.80 (0.90) | 0.90 (0.94) | **0.97** (0.99) |
| **Unseen repos** | AST | 0.36 (0.68) | 0.48 (0.80) | 0.80 (0.93) |
|  | AugAST | 0.59 (0.78) | 0.84 (0.92) | **0.92** (0.96) |

Table 3: Accuracy (and top–5 accuracy) on the Fill–In–The–Blank task, depending on which type of GNN the model uses. See Table 1 for explanations of the abbreviations. All models use AugAST as their code representation.

|  |  | GGNN | DTNN | RGCN |
|---|---|---|---|---|
| **Seen repos** | Closed Vocab | 0.80 (0.90) | 0.72 (0.84) | 0.80 (0.90) |
|  | GSC | **0.97** (0.99) | 0.89 (0.95) | 0.95 (0.98) |
| **Unseen repos** | Closed Vocab | 0.59 (0.78) | 0.46 (0.68) | 0.62 (0.79) |
|  | GSC | **0.92** (0.96) | 0.80 (0.89) | 0.88 (0.95) |

name (in the form of the sequence of words that compose the name). An example instance from our dataset is shown in Figure 3.

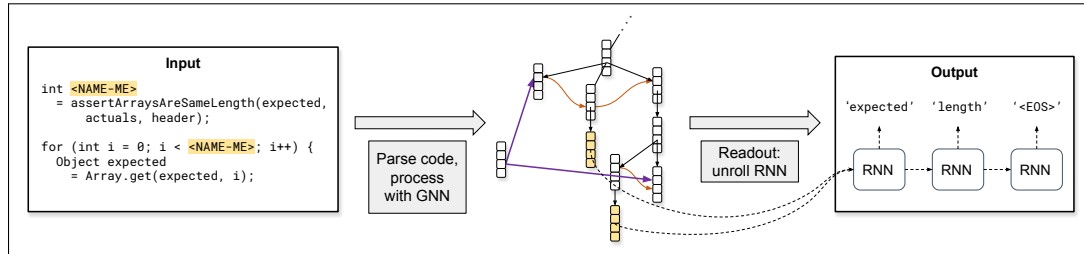

Figure 3: Example of a model's procedure for completing the Variable Naming task. Each Variable Naming instance is created by replacing all uses of some variable (`expectedLength`, in this example) with a special `<NAME-ME>` token. The model then processes the code as depicted in Figure 1. To produce outputs, the model takes the mean of the `<NAME-ME>` nodes' hidden states (depicted here in orange), uses them as the initial hidden state of a Recurrent Neural Network, and unrolls this RNN to produce a name as a sequence of words.

To produce a name from the output of the GNN, our models used the readout function of Allamanis et al. (2018). This readout function computes the mean of the hidden states of the `<NAME-ME>` nodes and passing it as the initial hidden state to a 1–layer Gated Recurrent Unit (GRU) RNN (Cho et al., 2014). This GRU is then unrolled to produce words in its predicted name, in the style of a traditional NLP decoder. We used a fixed length unrolling of 8 words, as 99.8% of names in our training set were 8 or fewer words long. The models were trained by cross entropy loss over the sequence of words in the name.

To decode each hidden state output of the GRU $h$ into a probability distribution $P_{\text{vocab}}(\mathbf{w}|h)$ over words $\mathbf{w}$, the Closed Vocab and CharCNN models pass $h$ through a linear layer and a softmax layer with output dimension equal to the number of words in their closed vocabularies (i.e. a traditional decoder output for NLP). In contrast, the GSC model not only has access to a fixed–size vocabulary

but can also produce words by pointing to cache nodes in its Graph–Structured Cache. Specifically, it uses a decoder architecture inspired by the Pointer Sentinel Mixture Model of Merity et al. (2017): the probability of a word $\mathbf{w}$ being the GSC decoder's output given that the GRU's hidden state was $\boldsymbol{h}$ is

$$P(\mathbf{w}|\boldsymbol{h}) = P_{\text{graph}}(\mathbf{s}|\boldsymbol{h})P_{\text{graph}}(\mathbf{w}|\boldsymbol{h}) + (1 - P_{\text{graph}}(\mathbf{s}|\boldsymbol{h}))P_{\text{vocab}}(\mathbf{w}|\boldsymbol{h})$$

where $P_{\text{graph}}(\cdot|\boldsymbol{h})$ is a conditional probability distribution over cache nodes in the GSC and the sentinel $\mathbf{s}$, and $P_{\text{vocab}}(\cdot|\boldsymbol{h})$ is a conditional probability distribution over words in a closed vocabulary. $P_{\text{graph}}(\cdot|\boldsymbol{h})$ is computed by passing the hidden states of all cache nodes and the sentinel node through a single linear layer and then computing the softmax dot–product attention of these values with $\boldsymbol{h}$. $P_{\text{vocab}}(\cdot|\boldsymbol{h})$ is computed as the softmax of a linear mapping of $\boldsymbol{h}$ to indices in a closed vocabulary, as in the Closed Vocab and CharCNN models. If there is no cache node for $\mathbf{w}$ in the Augmented AST or if $\mathbf{w}$ is not in the model's closed dictionary then $P_{\text{graph}}(\mathbf{w}|\boldsymbol{h})$ and $P_{\text{vocab}}(\mathbf{w}|\boldsymbol{h})$ are 0, respectively.

The performance of our GSC versus other methods is reported in Table 4. More granular performance statistics are reported in Appendix Table 6. We also compare the performance of different GNNs in Table 5.

Table 4: Accuracy on the Variable Naming task. Our model is the AugAST–GSC. The first number in each cell is the accuracy of the model, where we consider a correct output to be exact reproduction of the full name of the obfuscated variable (i.e. all the words in the name and then a `EOS` token). The second, parenthetical numbers are the top–5 accuracies, i.e. whether the correct full name was among the 5 most probable sequences output by the model. See Table 1 for explanations of the abbreviations. All models use Gated Graph Neural Networks as their GNN component.

|  |  | Closed Vocab | CharCNN | Pointer Sentinel | GSC |
|---|---|---|---|---|---|
| **Seen repos** | AST | 0.23 (0.31) | 0.22 (0.28) | 0.19 (0.33) | 0.49 (0.67) |
|  | AugAST | 0.19 (0.26) | 0.20 (0.27) | 0.26 (0.40) | **0.53** (0.69) |
| **Unseen repos** | AST | 0.05 (0.07) | 0.06 (0.09) | 0.06 (0.11) | 0.38 (0.53) |
|  | AugAST | 0.04 (0.07) | 0.06 (0.08) | 0.08 (0.14) | **0.41** (0.57) |

Table 5: Accuracy (and top–5 accuracy) on the Variable Naming task, depending on which type of GNN the model uses. See Table 1 for explanations of the abbreviations. All models use AugAST as their code representation.

|  |  | GGNN | DTNN | RGCN |
|---|---|---|---|---|
| **Seen repos** | Closed Vocab | 0.19 (0.26) | 0.23 (0.31) | 0.27 (0.34) |
|  | GSC | **0.53** (0.69) | 0.33 (0.48) | 0.46 (0.63) |
| **Unseen repos** | Closed Vocab | 0.04 (0.07) | 0.06 (0.08) | 0.06 (0.09) |
|  | GSC | **0.41** (0.57) | 0.25 (0.40) | 0.35 (0.49) |

## 6 DISCUSSION

As can be seen in Tables 2 and 4, the addition of a GSC improved performance on all tasks. Our full model, the AugAST–GSC model, outperforms the other models tested and does comparatively well at maintaining accuracy between the seen and unseen test repos on the Variable Naming task.

To some degree the improved performance from adding the GSC is unsurprising: its addition to a graph–based model is essentially just adding extra features and doesn't remove any information or flexibility. Under a satisfactory training regime, a model could simply learn to ignore it if it is unhelpful, so its inclusion should never hurt performance. The degree to which it helps, though, especially on the Variable Naming task, suggests that a GSC is well worth using for some tasks. Moreover, the fact that the Pointer Sentinel approach shown in Table 4 performs noticeably less well than the full GSC approach suggests that the relational aspect of the GSC is key: simply having the ability to output out–of–vocabulary words without relational information about their usage appears to be much less helpful.

The downsides of using a GSC are thus primarily computational. Our GSC models ran about 30% slower than the Closed Vocab models. Since we capped the graph size at 500 nodes, the slowdown is presumably due to the large number of edges to and from the graph cache nodes. Better support for sparse operations on GPU in deep learning frameworks would be useful for alleviating this downside.

In the near term, there remain a number of design choices to explore regarding AST– and GNN– models for processing source code. Adding information about word order to the GSC might improve performance, as might constructing the vocabulary out of subwords rather than words. It also might help to treat variable types as the GSC treats words: storing them in a GSC and connecting them with edges to the variables of those types; this could be particularly useful when working with code snippets rather than fully compilable code. For the Variable Naming task, there are also many architecture choices to be explored in how to produce a sequence of words for a name: how to unroll the RNN, what to use as the initial hidden state, etc.

In the longer term, given that all results above show that augmenting ASTs with data– and control– flow edges improves performance, it would be worth exploring other static analysis concepts from the Programming Language and Software Verification literatures and seeing whether they could be usefully incorporated into Augmented ASTs. Better understanding of how Graph Neural Networks learn is also crucial, since they are central to the performance of our model and many others. Additionally, the entire domain of machine learning on source code faces the practical issue that many of the best data for supervised learning on source code — things like high–quality code reviews, integration test results, code with high test coverage, etc. — are not available outside private organizations.

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

APPENDIX

ADDITIONAL EXPERIMENT INFORMATION

Table 6: Extra information about performance on the Variable Naming task. Entries in this table are of the form "subword accuracy, edit distance, edit distance divided by real name length". The edit distance is the mean of the character–wise Levenshtein distance between the produced name and the real name.

|  |  | Closed Vocab | CharCNN | Pointer Sentinel | GSC (ours) |
|---|---|---|---|---|---|
| **Seen repos** | AST | 0.30, 7.22, 0.94 | 0.28, 8.67, 1.08 | 0.32, 8.00, 1.07 | 0.56, 3.87, 0.39 |
|  | AugAST | 0.26, 7.64, 0.94 | 0.28, 7.46, 0.99 | 0.35, 6.51, 0.77 | 0.60, 3.68, 0.37 |
| **Unseen repos** | AST | 0.09, 8.66, 1.23 | 0.10, 8.82, 1.12 | 0.13, 9,39, 1.37 | 0.42, 4.81, 0.59 |
|  | AugAST | 0.09, 8.34, 1.14 | 0.10, 8.16, 1.12 | 0.14, 8.03, 1.07 | 0.48, 4.28, 0.49 |

DATASET INFORMATION

Table 7: Repositories used in experiments. All were taken from the Maven repository (https://mvnrepository.com/). Entries are in the form "group/repository name/version".

| **Seen Repos** |
|---|
| com.fasterxml.jackson.core/jackson-core/2.9.5 |
| com.h2database/h2/1.4.195 |
| javax.enterprise/cdi-api/2.0 |
| junit/junit/4.12 |
| mysql/mysql-connector-java/6.0.6 |
| org.apache.commons/commons-collections4/4.1 |
| org.apache.commons/commons-math3/3.6.1 |
| org.apache.commons/commons-pool2/2.5.0 |
| org.apache.maven/maven-project/2.2.1 |
| org.codehaus.plexus/plexus-utils/3.1.0 |
| org.eclipse.jetty/jetty-server/9.4.9.v20180320 |
| org.reflections/reflections/0.9.11 |
| org.scalacheck/scalacheck_2.12/1.13.5 |
| org.slf4j/slf4j-api/1.7.25 |
| org.slf4j/slf4j-log4j12/1.7.25 |
| **Unseen Repos** |
| org.javassist/javassist/3.22.0-GA |
| joda-time/joda-time/2.9.9 |
| org.mockito/mockito-core/2.17.0 |

MODEL INFORMATION

Table 8: Edge types used in Augmented ASTs. The initial AST is constructed using the `AST` and `NEXT_TOKEN` edges, and then the remaining edges are added. In other words, the the "AST" model from Table 1 uses a graph that contains only the `AST` and `NEXT_TOKEN` edge types (and `WORD_USE` if it also uses a GSC), while the "AugAST" model contains all the edge types below. The reversed version of every edge is also added as its own type (e.g. `reverse_AST`, `reverse_LAST_READ`) to let the GNN message passing occur in both directions.

| Edge Name | Description |
|---|---|
| `AST` | The edges used to construct the original AST. |
| `NEXT_TOKEN` | Edges added to the original AST that specify the left–to–right ordering of the children of a node in the AST. These edges are necessary since ASTs have ordered children, but we are representing the AST as a directed multigraph. |
| `COMPUTED_FROM` | Connects a node representing a variable on the left of an equality to those on the right. (E.g. edges from `y` to `x` and `z` to `x` in `x = y + z`.) The same as in Allamanis et al. (2018). |
| `LAST_READ` | Connects a node representing a usage of a variable to all nodes in the AST at which that variable's value could have been last read from memory. The same as in Allamanis et al. (2018). |
| `LAST_WRITE` | Connects a node representing a usage of a variable to all nodes in the AST at which that variable's value could have been last written to memory. The same as in Allamanis et al. (2018). |
| `RETURNS_TO` | Points a node in a return statement to the node containing the return type of the method. (E.g. `x` in `return x` gets an edge pointing to `int` in `public static int getX(x)`.) |
| `LAST_SCOPE_USE` | Connects a node representing a variable to the node representing the last time this variable's name was used in the text of the code (i.e. capturing information about the text, not the control flow), but only within lexical scope. This edge exists to try and give the non–GSC models as much lexical information as possible to make them as comparable with the GSC model. |
| `LAST_FIELD_LEX` | Connects a field access (e.g. `this.whatever` or `Foo.whatever`) node to the last use of this.whatever (or to the variable's initialization, if it's the first use). This is not lexical–scope aware (and, in fact, can't be in Java, in general). |
| `FIELD` | Points each node representing a field access (e.g. `this.whatever`) to the node where that field was declared. |
| `WORD_USE` | Points cache nodes to nodes representing variables in which the vocab word was used in the variable's name. |

