# OpenReview forum: "Open Vocabulary Learning on Source Code with a Graph-Structured Cache"
_ICLR.cc/2019/Conference_

### Official Review · AnonReviewer1 · 2018-10-31
**Improved graph representation for learning from programs**

**Rating:** 6
**Confidence:** 5

**Review:**

The submission presents an extension to the Allamanis et al ICLR'18 paper on learning from programs as graphs. The core contribution is the idea of introducing extra nodes and edges into the graph that correspond to (potentially rare) subwords used in the analyzed program code. Experiments show that this extended graph leads to better performance on two tasks, compared to a wide range of baseline methods.

Overall, this is a nice paper with a small, incremental idea and substantial experiments that show its practical value. I only have minor comments / questions on the actual core content. However, the contribution is very incremental and of interest to a specialized subsegment of the ICLR audience, so it may be appropriate to reject the paper and redirect the authors to a more specialized venue.

Minor comments:
- There's a bunch of places where \citep/\citet are mixed up (e.g., second to last paragraph of page 2). It would make sense to go through the paper one more time to clean this up.
- Sect. 4: I understand the need to introduce context, but it feels that more space should be spent on the actual contribution here (step 3). For example, it remains unclear why this extra nodes / edges are only introduced for subwords appearing in variables - why not also for field names / method names?
- Sect. 5: It would be helpful if the authors would explicitly handle the code duplication problem (Lopes et al., OOPSLA'17), or discuss how they avoided these problems. Duplicated data files occurring in several folds are a significant risk to the validity of their experimental findings, and very common in code corpora.
- Table 1: It is unclear to me what the "Pointer Sentinel" model can achieve. Without edges connecting the additional words to where they occur, it seems that this should not be performing different than "Closed Vocab", apart from noise introduced by additional nodes.
- Table 1: Do Pointer Sentinel/GSC use a CharCNN to embed node labels of nodes that are not part of the "cache", or a closed vocabulary? [i.e., what's the embedding of a variable "foo"?] If not, what is the performance of the GSC model with CharCNN-embeddings everywhere? That would be architecturally simpler than the split variant, and so may be of interest.
- Page 6: When truncating to 500 nodes per graph: How many graphs in your dataset are larger than that?
- Page 7: Do you really use attention over all nodes, instead of only nodes corresponding to variables? How do you deal with results where the model picks a non-variable (e.g., a corresponding cache node)? Does this happen?

---

> ### Author Response · Authors · 2018-11-07
> **Response to AnonReviewer1**
>
>
> Thanks very much for your time and comments, AnonReviewer1.
>
> Regarding your overall comment, we agree, though our hope is that the graph-structured cache strategy we propose will be of general use in open set/open vocabulary learning problems.  We simply focused on a particularly acute open vocabulary learning problem in this paper to explore its utility.  We will add this motivation more clearly to the paper to make it more relevant to a wider audience.
>
> Regarding the minor comments, thank you for your careful reading!  We will upload a version ASAP that addresses all of these, but to answer your questions:
> - Sect 4.: We actually do introduce them for field names and method names depending on the task - we will make that clearer.
> - Sec5 5.: We checked for duplicate code with CPD (https://pmd.github.io/latest/pmd_userdocs_cpd.html) and didn't find a worrying amount.  Out of the 500k nonempty, noncomment lines of code in our dataset, about 92k lines were duplicates of some other line in the dataset, with the majority of contiguous, duplicated lines of code containing fewer than 150 tokens and only being duplicated once.  We didn't find any duplicated files in our code.
> - Table 1: The Pointer Sentinel model can incorporate words from the vocabulary cache in its output by pointing to them with attention.  The Closed Vocab model can only produce names using words from its closed vocabulary.  So as you say, the only difference between the Pointer Sentinel model and our full model is the absence of the edges indicating word usage, but both are fairly different from the Closed Vocab model.
> - Table 1: Yes, Pointer Sentinel/GSC use a CharCNN to embed node labels for all non-internal nodes in the AST, including variables like "foo".
> - Page 6: About 53% are larger.
> - Page 7: We do.  If the model picks a non-variable, it is counted as a mistake.  But this (essentially) never happens: non-variable nodes are identified by their node type, so the model learns within a few batches not to attend to any non-variable nodes.

---

### Official Review · AnonReviewer2 · 2018-11-01
**A subword embedding model for codes. What's new?**

**Rating:** 4
**Confidence:** 4

**Review:**

The paper introduces a new way to use a subword embedding model 2 tasks related to codes: fill-in-blank and variable naming.

* pros:
- the paper is very well written.
- the model is easily to reimplement.
- the experiments are solid and the results are convincing.

* cons:
- the title is very misleading. In fact, what the paper does is to use a very shallow subword embedding method for names. This approach is widely used in NLP, especially in machine translation.
- the work is progressing, meaning that most of it is based on another work (i.e. Allamanis et al 2018).

* questions:
- how to build the (subword) vocabulary?

---

> ### Author Response · Authors · 2018-11-07
> **Response to AnonReviewer2**
>
>
> Thanks very much for your time and comments, AnonReviewer2.
>
> To respond to your title, first con, and question, I'm worried that a big misunderstanding has been caused somehow.  We don't use a "subword vocabulary" per se, and our embedding strategy is very ancillary to the main contribution of our work.  We use a CharCNN embedding in some of the models - is what you are referring to?  If so, it's very minor part to our main contribution, which is the usage of the graph structured cache.  (Hence our title.)  Were you referring to this cache as the shallow subword embedding?  Or have I misunderstood your comment?
>
> To respond to your second con, we are certainly continuing in the same direction as the excellent work in [Allamanis et al. 2018].  But our contributions extend quite a bit farther than that paper: we introduce an entirely new way of handling an open vocabulary, show that it improves performance on two well-studied tasks, present experiments with more Graph Neural Network architectures, and do all this on Java - a much more widely used language.  Would you consider this a fair characterization, or are we overstating the case?
>
> Thanks again!

---

> > ### Comment · AnonReviewer2 · 2018-11-08
> > **response**
> >
> > Thanks for the response. I would like to clarify my points centering around your main contribution:
> >
> > "3. Further augment the Augmented AST by adding a Graph–Structured Cache. That is, add a
> > node to the Augmented AST for each vocabulary word encountered in the input instance. Then
> > connect each such “cache node” with an edge (of edge type WORD USE) to all variables whose
> > names contain its word."
> >
> > Firstly, I think that the title is misleading because it's too much to claim that your vocabulary model uses "Graph–Structured Cache". Of course, most (or every) math objects can be represented by graphs. But here, the graph is too shallow. You have a layer of words and connect it to phrases (or variable/function names, which can be considered as phrases).
> >
> > Secondly, your model does remind me of subword approaches in NLP. For instance, I believe that you split variable/function names into tokens (such as "getGuavaDictionary" into "get", "Guava", "Dictionary"). Then
> > vector(name) = f(vector(tok_1),...,vector(tok_n)). If tok_k isn't in your voca, you use a charNN to combute vector(tok_k). If I understand it correctly, this method is widely used in NLP.
> >
> > If you think I misunderstood your model, I'm willing to change my review and I hope you will write your model in a clearer way.

---

> > > ### Author Response · Authors · 2018-11-08
> > > **Response to AnonReviewer2**
> > >
> > > Thank you so much for the detailed comments!  These are really helpful.
> > >
> > > Regarding 1: You're exactly right about the model structure, and I'm completely with you that "graph" is a term so flexible as to be often unhelpful.  We just had to pick a name for the model feature we were introducing, and we hoped "graph-structured cache" was clear and correct: it's a collection of words represented as nodes in a graph.
> > > But I entirely see how the name "graph-structured cache" might cause a reader expect to see a complicated adjacency structure within the cache nodes.  There is "depth" due to the message passing in the GNN, but I'll ask my coauthors and other readers if we can find a clearer name.
> > >
> > > Regarding 2: This is entirely our fault for not being clearer.
> > > In the Fixed Vocab baseline model, vector(name) = f(vector(word_1), ..., vector(word_n)).  (No CharCNN involved.  <unk> token used if word_k isn't in the vocabulary.)
> > > In the CharCNN baseline model, vector(name) = CharCNN(name).  (No splitting name into words.)
> > > But in our GSC model there is no single vector(name) exactly: a variable's name is "embedded" as CharCNN(name) along with edges connecting the variable to word nodes in graph-structured cache.  E.g. initializing a node containing a variable named "getGuavaDictionary" involves producing a vector CharCNN("getGuavaDictionary") and also adding "WORD_USE"/"reverse_WORD_USE" edges pointing to/from the "get", "guava" and "dictionary" nodes in the GSC, each of which contains CharCNN(word).
> > > So the baseline models are indeed standard NLP approaches, but ours (as far as we know) is new.  I'll edit the document to make this entirely clear.
> > >
> > > Thanks very much again for helping us improve our presentation!

---

### Official Review · AnonReviewer3 · 2018-11-01
**Overly complicated techniques for previously well-addressed tasks in literature**

**Rating:** 4
**Confidence:** 4

**Review:**

(updated with some summaries from discussion over the initial review)

The paper discusses the topics of predicting out-of-vocabulary tokens in programs abstract syntax trees. This could have application in code completion and more concretely two tasks are evaluated:
 - predicting a missing reference to a variable (called FillInTheBlank)
 - predicting a name of a variable (NameMe)

Unfortunately, the paper proposes overly complex and strange formulations of these tasks, heavy implementation with unnecessary (non-motivated) neural architectures and as a result, does not demonstrate state-of-the-art performance or precision on comparable tasks. Figure 1 shows the complexity of the approach, with multiple steps of building a graph, introducing the vocabulary cache to then produce a vector at every node of the input tree of the program (instead of creating architecture for a given task), yet simple analysis over which variables can be chosen is missing.

The FillInTheBlank task is badly defined already on the running example. The goal is to select a variable to fill in a blank and already in the example on Figure 2, one of the candidate variables is out of scope at the location to fill. The motivation for the proposed formulation with building a graph and then computing attention over nodes in that graph is unclear and experiments do not help it. For example, [1] (also cited in the paper) solves the same problem more cleanly by considering only the variables in the scope*. There is no good experimental comparison to that work, but it is unlikely it will perform worse. Also [1] does not suffer from vocabulary problems for that task.

Summary discussion below: the experiments here are incomparable on many levels with prior works: different architecture details, different even smaller dataset than from [1]. There is a third-party claim that on a full system, the general idea improves performance, but I take it with a grain of salt as no clean experiment was yet done. The reviewer notes that the authors disagree the baselines are not meaningful.

The NameMe tasks also shows the weakness of the proposed architectures. This work proposes to compute vectors at every node where a variable occurs and then to average them and decode the variable name to predict. In comparison, several prior works introduce one node per variable (not per occurrence), essentially removing the long distance relationships between occurrences of the same variable variables and removing the need to average vectors and enforcing the same name representation at every occurrence of the variable [name]. The setup here is incomparable to specialized naming prior works, one feature (a node per variable) is replaced with another (a node per subtoken), but for baselines authors choose to only to be similar to [1]. Also, while not on the same dataset, [2,3] consistently get higher accuracy on a related and more complicated task of predicting multiple names at the same time over multiple programming languages and with much simpler linear models. This is not surprising, because they propose simpler architectures better suited for the NameMe task.

[1] Miltiadis Allamanis, Marc Brockschmidt, and Mahmoud Khademi. Learning to represent programs
with graphs. ICLR 2017
[2] Veselin Raychev, Martin Vechev, and Andreas Krause. Predicting program properties from Big
Code
[3] Uri Alon, Meital Zilberstein, Omer Levy, Eran Yahav. A General Path-Based Representation for Predicting Program

* corrected text

---

> ### Author Response · Authors · 2018-11-07
> **Response to AnonReviewer3**
>
>
> Thanks very much for your time and comments, AnonReviewer3.
>
> For readability, let me list my responses and questions with references to your review:
>
> 1) "...strange formulations of these tasks, which are overly complex..."
> Did you mean the tasks themselves were overly complex, or our formulations of them were overly complex?  If the former, we should point out that we test on nearly identical tasks to [Allamanis et al. 2018].
>
> 2) "...does not demonstrate state-of-the-art performance..."
> We are not aware of any work that achieves better performance than ours without using >10x more data.  But even so, while models tailored to each task have certainly performed better, as far as we know we are the only model that uses the same pipeline (in fact, even the same hyperparameters) to achieve comparable to state-of-the-art performance on both these tasks on Java code.  Did you have a reference in mind that does similar?  If so, we want to make sure we cite it!
>
> 3) "Figure 1 shows the complexity of the approach,..."
> We were aiming for thoroughness with this figure, hence showing all the steps.  But compared to prior works, we only add one new step - we just depict the entire procedure in our figure.
>
> 4) "...introducing the vocabulary cache to then produce a vector at every node of the input tree of the program, which is unnecessary."
> Were you saying the cache was unnecessary, or vectorizing every node was unnecessary?
> If the former, we feel the experiments refute that, showing that the cache improved performance quite a bit.
> If the latter, it is certainly *possible* that there exists an architecture for which this is unnecessary, but this is what every Graph-Neural-Network-based approach does.  Doing so is not particularly computationally expensive.
>
> 5) "The FillInTheBlank task is badly defined already on the running example. ...one of the candidate variables is out of scope at the location to fill."
> This was intentional: we wanted our models to learn to consider lexical scope via the AST representation.  Indeed, as you say, we could likely improve performance by restricting the model's attention in a scope-aware way - but our objective was to compare architectures, not to maximize performance on this task.
> Thanks for pointing out the confusion: we will make this clearer in the paper.
>
> 6) "Also [1] does not suffer from vocabulary problems for that task."
> The Closed Vocab + AugAST entry in Table 2 is the same model as in [1].  So while it may not have suffered from vocabulary problems in [1]'s C# dataset, it indeed suffered from fairly significant vocabulary problems on our Java dataset.
>
> 7) "The NameMe tasks also shows the weakness of the proposed architectures. This work proposes to compute vectors at every node... In comparison, several prior works introduce one node per variable..."
> Were you suggesting here that the only weakness of our architecture was that we average several vectors on this task as opposed to using one vector?  Or were you giving one example of a more general critique?  If the latter, could you say more about what the general critique is?
>
> 8) "While not on the same dataset, [2,3] consistently get higher accuracy on a related and more complicated task..."
> Perhaps I'm misunderstanding, but I don't see how these papers' results are comparable to our results.
> [3] considers a JavaScript dataset, not a Java dataset.  These are very different languages, and the results of [2] suggest that variable naming in Javascript is significantly easier.
> [2] uses, as you say, a different dataset.  They use more than 16x as much data as we do and achieve around 5% better accuracy on this dataset (and only if you don't count method naming, which our model does).  But beyond their use of much more data, their model is designed specifically for the task of variable naming - ours is meant to be a general representation learning strategy for source code.  This is why we test it on two tasks and on entirely unseen repositories.
> Does this bear upon your concerns, or have I misunderstood your comment?
>
> Thanks again!

---

> > ### Comment · AnonReviewer3 · 2018-11-11
> > **Incorrect claims in rebuttal about prior works, please read the references.**
> >
> > First, there are comparisons for two tasks: FillInTheBlank vs NameMe.
> >
> > FillInTheBlank is addressed in
> > [1] Miltiadis Allamanis, Marc Brockschmidt, and Mahmoud Khademi. Learning to represent programs
> > with graphs. ICLR 2017
> >
> > 1). The formulation is unnecessarily complex. The final task is nearly identical to [1], but the means to solve it introduce problems that do not exist for this task.
> > 4) See 6. [1] solves VarMisuse without the need for any cache.
> > 6) VarMisuse from [1] does not suffer from vocabulary problems, because they pick from variables in scope and do softmax from their computed vectors. See also row "Node Labels: Tokens instead of subtokens" in their ablation study that there is no loss in accuracy from vocabulary. Vocabulary does not play a role here if the task is defined properly.
> > Also [1] does not have C++ dataset, only C# dataset.
> >
> > NameMe is task that is previously solved with other methods and architectures.
> > [2] Veselin Raychev, Martin Vechev, and Andreas Krause. Predicting program properties from Big
> > Code
> > [3] Uri Alon, Meital Zilberstein, Omer Levy, Eran Yahav. A General Path-Based Representation for Predicting Program
> > [2,3] are solving much more complex problem with higher precision. Have a look also at: https://arxiv.org/pdf/1809.05193.pdf , which uses neural networks.
> >
> > 1) See how the task is formulated in [2,3]. What is proposed in Figure 1 is strictly more complicated and NameMe solves only a subset of the task from [2,3]. Experimentally, there is code from these prior works to compare to if results are meant to become comparable
> > 2) This is a weakness of the paper, not a strength. Evaluation does not help conveying a message.
> > 8) Method names are also counted in [2,3]. Again, aiming to make the results incomparable to prior works certainly does not make the submission stronger.
> >
> > Ideally, the a selling point of this paper may be that one architecture is "good" in both of these tasks. However, it is  unlikely to be state-of-the-art for any of them and one architecture for the two tasks needs stronger application motivation.

---

> > > ### Author Response · Authors · 2018-11-11
> > > **Response to AnonReviewer3**
> > >
> > > Thanks again for the comments and clarifications, AnonReviewer3.
> > >
> > > We agree with all your descriptions of the prior works you mention.  We had read them all before publishing our paper.  But I'm afraid I don't see what "incorrect claims" we made - could you be more specific?
> > >
> > > We also certainly agree with your overarching point that our architecture, optimized as it is for performing multiple tasks, is unlikely to be state-of-the-art for any one of them.  If you don't feel that developing an architecture for learning representations of source code that are useful for multiple tasks is a worthwhile goal, then I doubt there's anything we can say to convince you our paper is meritorious.

---

> > > ### Public Comment · ~Miltiadis_Allamanis1 · 2018-11-12
> > > **Thoughts on the FillInTheBlank task**
> > >
> > > Hi,
> > > I've followed the discussion here and as one of the authors of [1] I would like to mention two points with respect to the "FillInTheBlank" task:
> > >
> > > * I don't feel that the formulation used by the authors is unnecessarily complex. In particular, I find the fact that they avoid the painful speculative analysis for each candidate variable, which we needed to do, very appealing.
> > >
> > > * Although it's true that VarMisuse doesn't suffer from a vocabulary problem, per se, the idea of connecting all nodes with the same subtokens to a single "supernode" is interesting. Based on a workshop presentation by the authors earlier this year, we did implement this trick within our existing VarMisuse code, which gave a +5% absolute performance increase, placing the VarMisuse accuracy to around 90%. In that sense, I find this a valuable idea.
> > >
> > > (full disclosure: I am aware of the identity of the authors but I have no conflict of interest with them)
> > >
> > > -Miltos

---

### Comment · Area_Chair1 · 2018-11-30
**A good example**

Thank you to the reviewers of this paper for engaging in discussion not just with the authors, but with one another, and providing substantial and detailed reviews. You are an excellent example for the community, and demonstrate the high standard according to which papers should be evaluated in ML conferences. Your efforts are deeply appreciated.

AC

---

### Meta-Review · Area_Chair1 · 2018-12-13
**Results do not justify method complexity**

**Confidence:** 4
**Recommendation:** Reject

**Metareview:**

This paper introduces fairly complex methods for dealing with OOV words in graphs representing source code, and aims to show that these improve over existing methods. The chief and valid concern raised by the reviewers was that the experiments had been changed so as to not allow proper comparison to prior work, or where comparison can be made. It is essential that a new method such as this be properly evaluated against existing benchmarks, under the same experimental conditions as presented in related literature. It seems that while the method is interesting, the empirical section of this paper needs reworking in order to be suitable for publication.